# Measuring Strong, Skillful, Good and Transpersonal Will: The development of the Multidimensional Will Scale

**Andrea Bonacchi** [1,2]*, **Georgia Marunic** [3], **Carlotta Tagliaferro** [3], **Rebecca Boschi** [4], **Chloe Lau** [5], **Francesca Chiesi** [3]

1 Centro Studi e Ricerca Synthesis, Associazione di Promozione Sociale Sul Sentiero, Florence, Italy, 2 Clinical Epidemiology Unit, Oncological Network, Prevention and Research Institute -ISPRO, Florence, Italy, 3 Department of Neuroscience, Psychology, Drug and Child's Health (NEUROFARBA), University of Florence, Florence, Italy, 4 School of Psychology, University of Florence, Florence, Italy, 5 Centre for Addiction and Mental Health, Toronto, Ontario, Canada

* andreabonacchi2016@gmail.com

## Abstract

### Background and objective

This cross-sectional study aimed to provide a scale to assess different aspects of the will based on Roberto Assagioli's theory.

### Methods and results

The scale development followed three steps. Step 1 focused on operationalizing the construct and developing the items. It was carried out through several phases of item generation and refinement, resulting in a pool of 38 items. At Step 2 we tested the psychometric properties of the initial 38-item scale with the goal of excluding the items that weakened the structural validity and reliability of the scale. Descriptive, internal consistency, and exploratory factor analyses statistics were computed on a large sample (Sample 1: $N$ = 587; age: $M$ = 21.55, $SD$ = 4.14, 66% female) and they led to a five-dimension model (*Strong*, *Skillful*, *Good toward Self and Other*, and *Transpersonal Will*) and the exclusion of 15 items. Analyses conducted at Step 3 on a different sample (Sample 2: $N$ = 683; age: $M$ = 34.09, $SD$ = 16.27, 54% female) allowed for further refinement of the scale. Confirmatory factor analysis conducted on the resulting 19-item scale showed a good fit for the five-factor model ($\chi^2$ (142) = 507.63, $p$< .001, $TLI$ = .91; $CFI$ = .93; $RMSEA$ = .06 [90%CI: .06–.07]), and evidence of its invariance across genders and ages was provided. Reliability indices (internal consistency and intraclass correlation coefficients) were adequate (ranging from .66 to .83) and correlations with measures of related constructs supported the external validity of the scale.

### Conclusion

This study provides researchers, therapists, and counselors with an efficient measurement tool to assess Assagioli's construct of will.

**Data Availability Statement:** All relevant data are within the manuscript and its Supporting Information files.

**Funding:** The authors received no specific funding for this work.

**Competing interests:** The authors have declared that no competing interests exist.

## Introduction

The will has been a subject of widespread appeal that has fascinated philosophers and scholars over time from different disciplines like social sciences, psychology, and neuroscience. Indeed, the significance of voluntary acts and behaviors enables scholars to understand the driving force behind people's choices and commitment to action. Over time, numerous attempts have been made to define and discern between different conceptualizations of the will, such as will-power [1], ego strength [2], self-regulation and action control [3], self-control [4]. Alongside these theoretical approaches, in the second half of the last century, the Italian psychiatrist Roberto Assagioli developed a broad conceptualization of the will. In his work "The Act of Will" [5], he describes it as the psychological ability through which we regulate the expression of motivational drives (instincts, needs, values, emotions, etc.) in actions, attitudes, and behaviors. The will arises when people are facing an obstacle and enables them to confront it decisively, but also during periods of calm when there is a need to think over and make a decision.

According to Assagioli's theory, a fully developed will has some fundamental features: the Strong Will, the Skillful Will, the Good Will, and the Transpersonal Will. These aspects are linked to self-regulatory and action-control functions, particularly the first three which are related to the Personal Self. Every aspect of the will can be trained and Assagioli proposes a series of practices for this purpose. Specifically, Strong Will refers to a physical or mental effort made to overcome a hurdle or to reach a goal, and it drives people to act and persist. Skillful Will concerns the ability to achieve desired results with minimal effort, the capacity to develop the most effective strategy, to foresee the consequences of one's actions, and is based on knowledge and control of oneself and the principles underlying one's psychological functioning. Good Will is necessary to direct actions for the sake of good and it includes regulating and selecting goals consistent with one's well-being but also acting with empathy and prosocial behaviors to achieve right and balanced outcomes. The Transpersonal Will concerns the union with a higher self and needs that transcend the material world (i.e., the need to feel in unification with a higher force that completes one's existence), and it may be related to a person's spirituality or religious beliefs.

Linked to the different theories of volition that have been proposed [e.g., 1–4], different assessment scales were developed (e.g., the *Volition Components Inventory* [6] and the *Self-Control Scale* [7]). Since, to the best of our knowledge, there are no validated instruments to measure the will according to Assagioli's model [5], the present study aims to develop a tool to assess the above-described conceptualization of will in line with the growing need to provide operationalized and empirically supported definitions of constructs from psychodynamic, humanistic, and transpersonal psychology [8–10]. Moreover, considering that the will could be trained and strengthened, the development of a quantitative measure can be helpful in clinical and counseling interventions. Starting from the operationalization of Assagioli's theoretical conceptualization of the will and its aspects, several items were developed (Step 1). Investigating the psychometric properties of each item and the whole scale, the most suitable items were selected to maximize the reliability and validity of the instrument (Step 2) and then a second study was conducted to confirm the psychometric properties of the resulting scale (Step 3) that was named the *Multidimensional Will Scale* (MWS).

## Step 1: Initial development of the MWS

The scale development consisted of the following phases.

### Operational definition and item formulation

The starting point for the development of the MWS was to provide an operational definition of Assagioli's theoretical conceptualization of the will and its facets. The research team

involved five experts with training in psychosynthesis from the Italian Institute of Psychosynthesis. A good definition of the construct and its components ensures that the items can adequately reflect them [11]. Therefore, with the help of content area experts, a brief description of the five aspects of will based on Assagioli's [5] theorization was prepared and discussed during group meetings taking into account the potential use of the measure. Experts shared ideas on the type of item content that best represents each will facets. All experts agreed on splitting the *Good Will* aspect into *Good Will toward Ourselves* and *Good Will toward Others*. Assagioli emphasizes the importance of Good Will for the well-being of others as well as our own. To correctly display these two not mutually excluding facets, it was decided to consider them as two separate dimensions.

Thus, the operationalizations for each aspect of the will were agreed upon as follows. *Strong Will* is the impetus that drives people to act and makes them determined to persist and can be conceptualized as the physical or mental effort made to overcome a hurdle or to reach a goal. *Skillful Will* is the ability to achieve desired results by developing the most effective strategies, foreseeing the consequences of one's actions, reducing one's efforts, and using knowledge and control of the principles underlying one's psychological functioning. *Good Will toward Self* is required to regulate and select goals consistent with one's well-being and to be guided by self-love, while *Good Will toward Others* is the will acted with brotherly and altruistic love and requires empathy and prosociality. Thus, it contributes to achieving a suitable outcome for oneself considering, at the same time, the impact of the implemented actions on other people. Finally, *Transpersonal Will* concerns the union with a higher self and needs that transcend the material world, and it may be related to a person's spirituality, but not necessarily to religious issues.

Each expert was asked to submit, in written form, a list of at least five items per dimension. Then, a group discussion was made to select the item that better represented Assagioli's definition of the will, to merge similar items or choose the best one among them. Eventually, 50 items were retained.

## Item refinement and response format choice

Three psychology academics trained in psychosynthesis who were not involved in the item generation were asked to examine the 50-item list. Seven items deemed ineffective, vague, or lengthy were excluded. Five items considered too similar were merged or rephrased to give an effective formulation to each one of them [12]. The 38 selected items were again evaluated for clarity and appropriateness concerning each of the five starting dimensions. Items were deemed adequate. The instruction section was formulated and the response format chosen. Following Cox's [13] recommendations, the five-option response scale (from 1 = *Never* to 5 = *Always*) was opted for as it offers sufficient variety without overloading respondents with excessive response options.

## Content validity

The preliminary 38-item version of the MWS was presented to 10 experts in psychosynthesis from the Italian Institute of Psychosynthesis. They were introduced to the new operationalization of the will construct and the general aim of the study, and then asked to rate the item relevance on a 3-point scale (from *1 = no relevance* to *3 = high relevance*) and to assign them the dimension to which they believed each item pertained. Moreover, for each item, a space was left for comments or suggestions. This type of procedure is followed to ensure content validity [14]. The criterion adopted for item deletion was receiving an-average rating <2 and an assignment to the supposed dimension in less than 75% of the evaluations. For all items, the

relevance mean value was above the cut-off level and the correct dimension was identified in 87.5% to 100% of the cases. Finally, the experts' notes suggested minor adjustment to the item wording. The resulting MWS consisted of 38 items rated on a 5-point Likert scale and referring to five dimensions: 7 items were developed to assess the *Strong Will* (e.g., "*Obstacles motivate me to do more.*"), 6 items the *Skillful Will* (e.g., "*Before doing something important I think about its consequences over time.*"), 11 items the *Good Will toward Self* (e.g., "*I am committed to protecting and caring for myself.*"), 8 items the *Good Will toward Others* (e.g., "*When I act, I keep in mind the welfare of others.*"), and 6 items in the *Transpersonal Will* (e.g., "*In my choices I take into account profound and spiritual values.*"). The original item pool is reported in Appendix A.

## Step 2: In-between phases of the development of the MWS

The present study tested the psychometric properties of the items and the entire scale with the aim of consolidating the reliability and structural validity of the MWS by excluding some items.

### Method

**Participants.** The instrument was administered to 587 Italian adults ages 18 to 61 years ($M_{age}$ = 21.55, $SD$ = 4.14). Approximately 66% of participants identified as cisgender women, 33% as cisgender men, and 1% preferred not to say. The majority were students (91%) from a large university in central Italy.

The sample size was established referring to the requirements to develop a scale through factor analysis that depends on the number and type of the items, the extent of correlations between items in the population, average commonality, the number of factors, and the stability of the factor structure [15]. At this initial stage this information was unknown except for the initial number of items and the hypothetical 5-factor model derived by the theoretical background. Thus, as suggested by the literature as a rule of thumb, a minimum 500 or more cases can assure the stability, reliability, and replicability of a factor solution [16,17].

The study received the approval from the Ethics Committee for Research of the University of Florence as a part of a larger research project (Opinion No. 31, July 23, 2018). The participants were recruited, provided with study information, and asked to sign a written consent form. All participants took part in the study anonymously and voluntarily as no incentives were offered. The criteria for ineligibility were minor age and lack of informed consent.

**Measure and procedure.** Data were collected in paper and pencil format. Participants were asked to fill in a brief socio-demographic form and the 38-item version of the MWS described above. To investigate the temporal stability of the scale, a subsample of participants ($N$ = 228; 39% of the original sample) completed the MWS again in a four-to-five-week interval. This time interval was deemed appropriate because the measure refers to trait-like attributes. Since a trait has high stability, no significant changes were expected during this period [18].

**Analyses.** Prior to conducting the analyses, the missing values in the data were examined. For each item, the percentage of missing answers was calculated to check that they did not exceed 10% of the total answers. Listwise deletion was performed when there were >10% of missing answers; otherwise, the arithmetic mean of each item replaced the missing data [19]. All data analyses were performed on JASP for Windows software, version 0.16.4 [20].

The psychometric properties of the items were assessed by computing descriptive statistics (mean scores, standard deviations, skewness, and kurtosis). When developing a new measure, a key feature of the items is to produce as much score variance as possible, ideally assuming a

normal distribution form. Accordingly, the ones with skewness and kurtosis values below 1 were selected, as items closer to 0 are more likely to have a normal distribution, the basic assumption for most statistical analyses [21]. Furthermore, items closer to the mean value were kept as they convey more information and avoid respondents with a tendency to select endpoint rates [11].

Additionally, a series of Exploratory Factor Analyses (EFA) were carried out until a satisfactory structure was reached. To assess the suitability of the data for factor analysis, the *Kaiser-Meyer-Olkin* (KMO) test and Bartlett's test of Sphericity were computed. KMO values above .80 and a significant Bartlett test prove sampling adequacy. To determine the number of factors to retain, we performed a principal component (PC)-based parallel analysis (i.e., we selected factors with their eigenvalue greater than the parallel average random eigenvalue) together with the scree plot derived from the EFA based on principal axis factoring (PAF) extraction method. When factor analysis is supported by a theoretical model as in the present study, the combination of the two methods guarantees an optimal approach to factor selection [22] and it allows for avoiding the overestimation of the number of factors [23]. Additionally, while parallel analysis based on PC is used to reduce the number of variables by maximally preserving information from the original dataset [24], EFA based on PAF is preferred for the representation of the underlying latent construct [25]. The minimum item loading of .30 was considered for item retention [26] and for identifying cross-loadings [23]. Thus, the items with higher factor loadings were selected, and those with cross-loadings were excluded from the questionnaire. Then, the EFA with a fixed number of factors was performed several times to select items with greater loadings for each factor and determine the more suitable number of items for each factor.

The reliability of each dimension was assessed using Cronbach's Alpha ($\alpha$) and McDonald's Omega ($\omega$) coefficients for internal consistency with a relative 95% confidence interval. Values below .70 are considered unacceptable (.70 $\leq \alpha$ .79 fair; .80 $\leq \alpha \leq$ .89 good; $\alpha \geq$ .90 excellent) [27,28]. To determine the item's contribution to internal consistency, Alpha and Omega values were taken if the item dropped, while the item-total correlation value was taken as a measure of informativeness. The combination of these two indicators guided the choice of item deletion.

Thus, item selection was conducted by excluding items with poor variability, items that do not clearly contribute to measuring the construct dimensions, and to the reliability of the scale. After the item selection, EFA was repeated followed by internal consistency analyses for each dimension of MWS. The reliability was assessed also by test-retest analysis. Specifically, the intraclass correlation coefficient (ICC) was computed using a two-way mixed model based on absolute agreement with a single measure. Values less than .60 are indicative of poor reliability (.60 $\leq$ ICC $\leq$ .75 moderate, .75 $\leq$ ICC $\leq$ .90 good, and ICC $\geq$ .90 excellent) [29].

## Results

After examining the missing values, no cases were deleted because of more than 10% of missing responses. For each item, missing responses ranged from 0 to 6 (0% - 1%), thus they were replaced by the mean value of the respective item. Then, item selection and scale refinement followed several phases based on descriptives, exploratory factor analysis, and reliability indices.

**Item descriptive.** All response rates (range 1–5) were selected. The lowest mean item scores were obtained on IT7 ($M$ = 1.91, $SD$ = 1.11), and IT16 ($M$ = 1.78, $SD$ = .96). The absolute value of skewness ranged from .01 to 1.25, and from .01 to 1.10 for kurtosis, suggesting that IT7 ($Sk$ = 1.12), IT16 ($Sk$ = 1.25, $Ku$ = 1.06), and IT27 ($Ku$ = -1.10) deviated from

normality. Thus, the preliminary analysis produced the drop of IT7 (*"I would like to transcend my human limitation through union with someone, something greater and higher"*), IT16 (*"I perceive a contrast between my will and a greater will of a spiritual type"*), and 1T27 (*"I recognize that there is a will higher than the personal will, however, it is called: destiny, providence, etc."*). Detailed descriptive statistics are available in Supporting information (**S1A Table**).

**Exploratory Factor Analysis (EFA).** The first EFA was conducted on the 35 items of the MWS. The KMO = .82 indicated that the strength of the relationships between the items was fair. Bartlett's test of sphericity was significant (Bartlett's $\chi^2 (df = 595) = 7375.04$, $p < .001$), and indicated acceptability to proceed with the analysis. Consistently with the theoretical model, results from parallel analysis suggested five components to be retained, whose eigenvalues ranged from 6.11 to 1.67. Scree test analysis confirmed a five-component solution (see **Fig 1**) which accounted for 40% of the total variance. Four items (IT18, IT21, IT29, and IT31) showed loadings lower than .30 and were deleted. Thus, the factor analyses for a 31-item solution and with the number of factors fixed at five was repeated. From an item-selection perspective and based on Tabachnick and Fidell [21], IT12 has been eliminated given the limit loading on the respective factor (.31). All the items loaded on the expected factor (i.e., *Strong Will*, *Good Will toward Other*, *Good Will toward Self*; *Transpersonal Will*; *Skillful Will*) with two exceptions. Indeed, items IT22 and IT32 initially expected to belong to the *Skillful Will* dimension, loaded on the *Strong Will*. Hence, IT12, IT18, IT21, IT29, and IT31 were deleted, and the same factor analysis was repeated on the 30-item version. The following solution was obtained: *Strong Will* factor consisted of 9 items (loadings .39– .74), *Skillful Will* of 5 items (loadings .33 –.69), *Good Will toward Self* of 5 items (loadings .42 – .83), *Good Will toward Other* of 8 items (loadings .36 – .81), and *Transpersonal Will* of 3 items (loadings .81 – .84). Again, the same two items, IT22 and IT32, loaded on the *Strong Will* factor instead on the *Skillful Will* one. Thus, although these two items were initially developed to assess the *Skillful Will* dimension, the factor analyses repeatedly suggested that they load to the *Strong Will* dimension.

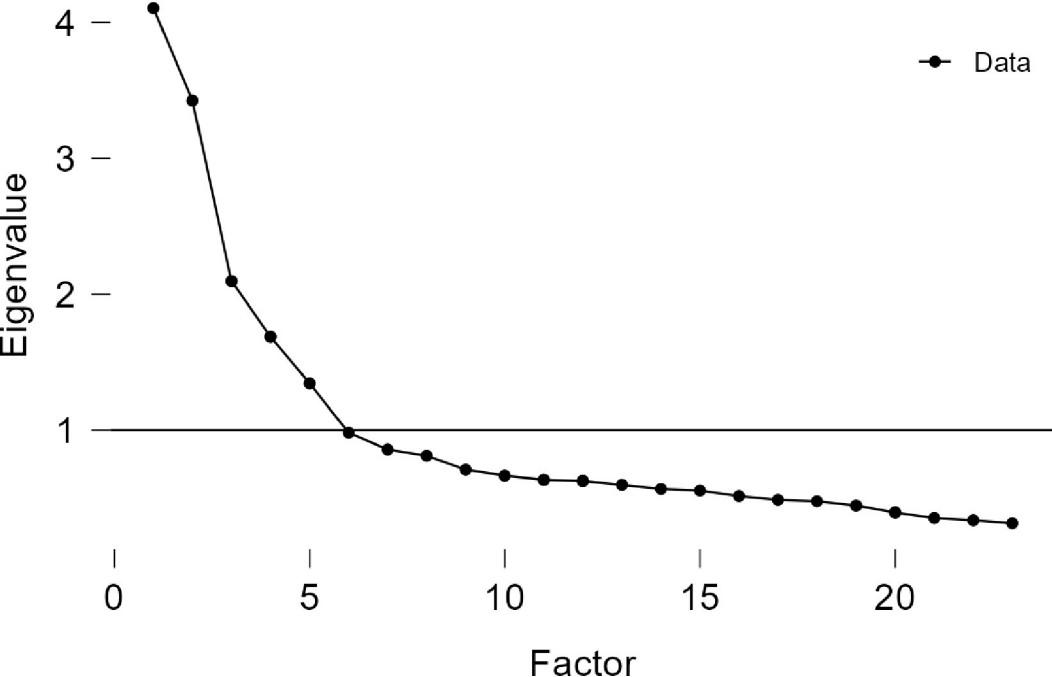

**Fig 1. Scree plot for 23-item solution of Multidimensional Will Scale (MWS).**

**Reliability.**    Cronbach's Alpha and McDonald's Omega coefficients confirmed fair to good levels of internal consistency for all dimensions but *Skillful Will* which was close to the acceptable threshold (item loadings on respective factors and internal consistency for the 30-item scale solution are presented in detail in Supporting information **S1B Table**). By analyzing the item-total correlations and assessing the Alpha and Omega levels when items deleted, it was observed that all items with a correlation lower than .40 did not contribute to internal consistency and were therefore dropped. Specifically, IT20 ($r = .38$; "*I like to lead others and take on roles of responsibility*"), IT28 ($r = .39$; "*I am able to realistically assess the possibilities of achieving my goals*") and IT37 ($r =. 33$; "*I am influenced by the opinion of others*") for the *Strong Will*. The IT1 ($r = .34$; "*When I have to make a choice, I pause to understand the point of view of others*"), IT 8 ($r = .31$; "*I try to act with respect for the natural environment*") and IT11 ($r =. 33$; "*I prefer to cooperate with others even if it slows me down in achieving my goals*") for the *Good Will toward Others*. The IT38 ($r = .32$; "*When I have to do something, I pause to think about the easiest and most practical way to do it*") was eliminated for the *Skillful Will*. The only exceptions for item-total correlation values were observed for IT9 ($r = .37$; "*I patiently look for the best way to do things*") and IT10 ($r = .38$; "*Doing something good for me makes me feel guilty*") as eliminating them would have decreased Alpha levels.

**EFA and reliability of the resulting 23-item version.**    Following the item selection analyses, fifteen items were eliminated and a 23-item version of the MWS comprised of five dimensions has been obtained (*Strong Will*: 6 items, *Skillful Will*: 4 items, *Good Will toward Self*: 5 items, *Good Will toward Other*: 5 items, and *Transpersonal Will*: 3 items). Thus, the EFA (Principal Axis Factoring, Oblimin rotation) was repeated with 23 items. The preliminary analyses indicated the acceptability of data to proceed with EFA (KMO = .81; Bartlett's $\chi^2(df = 253) = 5143.39$, $p < .001$)]. The scree plot confirmed the 5-factor solution (see **Fig 2**) which explained 50% of the total variance. All items had the appropriate loadings on their respective dimension ranging from .33 to .85 in absolute values. Detailed statistics are available in the Supporting information (**S1C Table**). The correlations between the factors reveals the relationships between *Strong Will* and *Good Will toward Self* ($r = .34$), *Good Will toward Others* with *Skillful*

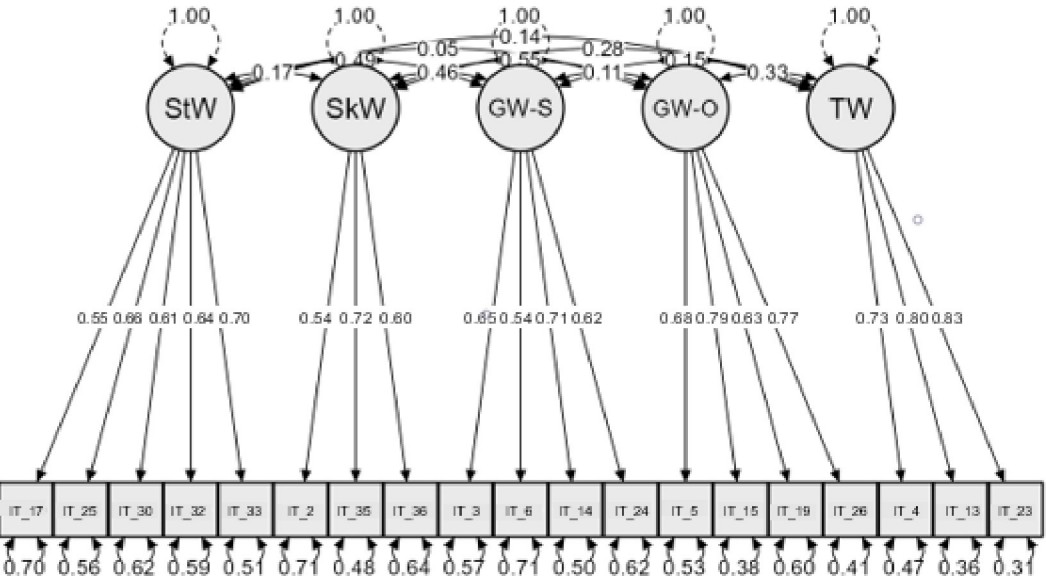

**Fig 2. Factor model of the 19-item Multidimensional Will Scale (MWS).** *Notes.* StW = Strong Will; SkW = Skillful Will; GW-S = Good Will toward Self; GW-O = Good Will toward Others; TW = Transpersonal Will.

*Will* (*r* = .25), and *Transpersonal Will* (*r* = .35). The remaining correlations between factors were not significant and ranged from .04 to .18.

Internal consistency measured with Cronbach's Alpha and McDonald's Omega coefficients for single dimensions were adequate to good for all dimensions and remained under the adequacy threshold for *Skillful Will*. However, the mean inter-item correlation was .32, a value deemed optimal [30]. Items are well within that range for the *Skillful Will* factor. For all dimensions, all items had an item-total correlation above .40, except for item 10 (*r* = .38), which was not, however, eliminated because of its contribution to internal consistency.

The ICCs ranged from good to excellent (.67–.95). Specifically, *Strong Will* = .67 [95% CI: .43–.81, *F* = 3.082 (*df1* = 53, *df2* = 53), *p* < .001], *Skillful Will* = .87 [95% CI: .74–.93, *F* = 9.33 (*df1* = 52, *df2* = 52), *p* < .001], *Good Will toward Self* = .71 [95% CI: .50–.84, *F* = 3.458 (*df1* = 51, *df2* = 51), *p* < .001], *Good Will toward Others* = .90 [95% CI: .82–.95, *F* = 11.591 (*df1* = 53, *df2* = 53), *p* < .001], and *Transpersonal Will* = .95 [95% CI: .91–.97, *F* = 18.66 (*df1* = 52, *df2* = 52), *p* < .001]. These findings provide good test-retest reliability evidence for the MWS.

## Discussion

The Step 2 allowed to test item properties and explore the factor structure of the scale. In line with Assagioli's theoretical framework, a 5-factor structure was observed and items saturated on the expected dimensions except for two items of the *Skillful Will* dimension (IT22 "*I recognize and use the inner qualities that can be useful to me in the pursuit of a goal.*"and IT32 "*I am good at finding solutions to overcome obstacles.*"). Thus, it was decided to reassign them to the Strong Will. Indeed, referring to efforts made to overcome a hurdle or to reach a goal, they were consistent with this will facet. The reliability indices confirmed the appropriateness of the selected items for each dimension.

These in-between phases of the scale development led to a 23-items version assessing the five dimensions of *Will* (*Strong*, *Wise*, *Good toward Self*, *Good toward Others*, and *Transpersonal*).

## Step 3: The final version of the MWS

The goal of Step 3 of the MWS development was threefold. First, we aimed at confirming the five-factor structure of the 23-item MWS in a different sample. Specifically, we employed a sample more representative of the general population since in Step 2 participants were mainly female university students. As the second aim, gender and age measurement invariance were assessed to ensure the scale is metrically equivalent between male and female respondents, and among different age groups respondents. If so, unbiased group comparisons can be made [31]. The third aim was to assess the validity of the MWS. Since Assagioli's model has not been studied from a psychometric point of view, the validity study had an explorative nature. Starting from the initial analysis of the single aspects of the will as conceived by Assagioli, it was possible to hypothesize some relations between the different aspects of the will and other constructs, which have been previously studied in relation to volition, to provide evidence of the construct validity. More precisely, if the correlations went in the expected direction, it may provide convergent and discriminant validity for the constructs defined by Assagioli. First, all will dimensions were expected to be related to the sense of mastery (i.e., the extent to which individuals perceive their life to be under their control) and self-control (i.e., the individual's disposition to correct one's behaviors, thoughts, and emotions to avoid undesirable outcomes), and seriousness (i.e., an attitudinal and habitual facet of a sober, pensive, and thoughtful frame of mind linked to the tendency to plan ahead of time and set long-range goals, and the tendency to prefer concrete- and rational-reasoned activities; [32]). Moreover, the relationships with resilience (i.e., the ability to cope with adversities) and sense of coherence (i.e., a personal

resource that relates to the ability to understand, give meaning to, and manage stressful life events) were explored.

## Method

**Participants.**   The questionnaire was administered to a sample (*N* = 683) aged between 18 and 84 years (*M* = 34.09, *SD* = 16.27), of which 54% were female. Ethical approval was granted by previous mentioned opinion of the Ethics Committee for Research at the University of Florence (Opinion n. 31 from 7/23/2018). Sampling was based on the "snowball" method [33] in which undergraduate students in a psychology course were invited to participate in an online questionnaire study and were also encouraged to recruit their acquaintances and relatives to participate. All participants were required to provide a written informed consent before questionnaire administration which took approximately 20 minutes to be completed. The criteria for ineligibility were minor age and lack of informed consent.

**Measures.**   Data were collected through an online questionnaire containing the 23-item MWS and other measures described hereafter. Control questions were included in the questionnaire to avoid response bias.

*Life Orientation Test-Revised* (LOT-R) [34], Italian version [35] was chosen to assess individual differences in dispositional optimism, defined as a generalized expectation of positive future outcomes. The scale consists of 6 items assessing optimism and pessimism (e.g., "*In times of uncertainty, I usually expect the best")* and 4 filler items (e.g., "*I get along very well with my friends*") measured on a five-point Likert scale (from 1 = *strongly disagree* to 5 = *strongly agree*). Higher scores are indicators of greater dispositional optimism. In the current sample, the scale has good internal consistency (α = .80).

*Sense of Coherence Scale-Revised* (SOC-R) [36], Italian version [37] was administered to assess the sense of coherence, defined as a personal resource that relates to the ability to understand, give meaning to, and cope with stressful life events. The scale consists of 13 items formulated as statements about life and comprises three dimensions: *Manageability* (5 items; e.g., "*One can always find a way to cope with painful things in life*"), *Balance* (4 items; e.g., "*I am convinced that a lot of negative feelings (e.g. rage) also have positive sides*"), and *Reflection* (4 items; e.g., "*Normally I can consider a situation from various perspectives*") rated on a five-point Likert scale (1 = *not at all true* to 5 = *extremely true*). Overall and single subscale scores are calculated. Higher scores are indicative of the greater presence of the measured trait. In the current sample, overall internal consistency is adequate (α = .70), for *Manageability* (α = .56), *Balance* (α = .50), and *Reflection* (α = .78).

*Pearlin-Schooler Mastery Scale* (PSMS) [38] assesses the sense of mastery, conceptualized as a coping mechanism to buffer the relationship between stressful events and psychological distress. It regards the extent to which individuals perceive life to be under their control. The scale consists of 7 items (e.g., "*What happens to me in the future depends mainly on me*") measured on a five-point Likert scale (from 1 = *strongly disagree* to 5 = *strongly agree*). Higher scores indicate a greater sense of mastery. The scale displayed good internal consistency in the current sample (α = .78).

*Connor-Davidson Resilience Scale* (CD-RISC-10) [39], Italian version [40] is a commonly used measure of resilience, defined as the ability to cope with adversities. The scale consists of ten items (e.g., "*I am able to cope with any obstacle in life*") measured on a five-point Likert scale (from 0 = *not true at all* to 4 = *true nearly all the time*). A higher score indicates greater resilience. The scale displays good internal consistency in the current sample (α = .86).

*Brief Self-Control Scale* (BSCS) [41], Italian version [42]. The construct measured by this scale relates to the individual's disposition to modify dominant responses (e.g., adjusting one's

behaviors, thoughts, and emotions) to avoid inappropriate or undesirable behaviors (e.g., those that produce strong immediate rewards) that are difficult to change or overcome. Italian adaptation of the scale consists of 7 items (e.g., "*I am good at resisting temptation*") measured on a five-point Likert scale (from 1 = *not at all like me* to 5 = *very much like me*). Higher scores indicate greater self-control. The scale displays adequate overall internal consistency in the current sample ($\alpha$ = .71), and lower for the two subscales: Impulse Control ($\alpha$ = .67) and *Self-Discipline* ($\alpha$ = .57) which, however, consist of just a few items.

*State-Trait Cheerfulness Inventory—Trait Version* (STCI-T60) [32], Italian version [43] is a multidimensional measure that assesses latent traits of cheerfulness, bad mood, and seriousness. Research has shown that these traits are linked with psychological well-being, positive affect, and emotion regulation. The scale consists of 60 items that provide scores on three factors: Cheerfulness (e.g., "*I can laugh easily*"), Seriousness (e.g., "*I prefer conversations that deal with important things and that are very deep*"), and Bad Mood (e.g., "*Compared to others, I can be really grumpy and grouchy*") measured on a four-point Likert scale (from 1 = *strongly disagree* to 4 = *strongly agree*). Higher scores indicate a higher temperamental basis of measured traits. In the current sample internal consistency was excellent for Cheerfulness ($\alpha$ = .93) and Bad Mood ($\alpha$ = .92), and good for Seriousness ($\alpha$ = .82)

**Analyses.** Prior to conducting the analyses, the missing values in the data were examined (see Study 1). All analyses were performed with JASP for Windows software, version 0.16.4 [20].

To verify the factorial structure of the MWS, Confirmatory Factor Analysis (CFA) was conducted using a diagonally weighted least squares (DWLS) estimation method. The data measured with Likert-type ratings are of ordinal nature and thus, it is preferable to use DWLS [44]. Furthermore, DWLS is preferred for large samples [45]. To assess the model fit following fit indices were evaluated: Comparative Fit Index (CFI), Tucker-Lewis index (TLI), and Root Mean Square Error of Approximation (RMSEA). Specifically, RMSEA values lower than .08 would suggest an adequate model fit, and CFI and TLI values in the range of .90 and .95 would suggest a moderate to excellent model fit [46,47].

Confirmatory Multigroup Factor Analysis (MGCFA) was used to test the invariance of the MWS between gender and age groups: female respondents ($N$ = 371; age range 18 to 84 years, $M_{age}$ = 33.92, $SD_{age}$ = 16.22) and male respondents ($N$ = 312; age range from 18 to 84 years $M_{age}$ = 34.29, $SD_{age}$ = 16.34), younger people ($N$ = 312; age range 18 to 29 years $M_{age}$ = 21.65, $SD_{age}$ = 2.09, 52.8% female) and older people ($N$ = 371; age range from 30 to 84 years $M_{age}$ = 52.99, $SD_{age}$ = 8.15, 56.5% female). Specifically, it was tested if the factor structure was consistent between genders and ages at the configural level (i.e., the construct is associated with the same set of items in each group), metric level (i.e., the relationships between the construct and items are not significantly different between group variables), scalar level (i.e., both factor coefficients and intercepts are equal between groups), and strict level (i.e., error terms do not differ between groups). Therefore, a series of hierarchically nested MGCFAs was applied. An unconstrained model (Model 0) was used to test for configural invariance. Subsequently, three more restrictive models were tested: Model 1 in which factor loadings were constrained to be equal across groups to test metric invariance, Model 2 in which factor loadings and intercepts were constrained to be equal across groups to test scalar invariance, Model 3 in which factor loadings, intercepts, and error terms were constrained to be equal across groups to test strict invariance. Each subsequent restriction was only applied if the previous was allowed. Comparisons were made between the last fitting model and the next most restricted one [48,49]. The models were compared using the chi-square-based likelihood ratio difference ($\Delta\chi^2$), Comparative Fit Index difference ($\Delta$CFI), and the Root Mean Square Error of Approximation difference ($\Delta$RMSEA). A significant value of $\Delta\chi^2$ together with a value of $\Delta$CFI $\Delta$RMSEA $\leq$.01 would

indicate invariance [48,50,51]. Whereas strict invariance is not required, scalar invariance is necessary to affirm that differences in scores are not affected by a measurement bias [52].

The reliability of the scale measured as internal consistency was assessed for each of the five dimensions by calculating Cronbach's Alpha ($\alpha$) and McDonald's Omega ($\omega$) coefficients.

Finally, Bayesian statistical analyses were used to evaluate the relationships among each dimension of the MWS and the variables in the study. Jeffreys' Bayes Factor describes the observed data using a priori and posterior distribution [53] which allows the quantification of evidence in favor of the alternative and null hypotheses [54]. Bayes Factors for evidence of alternative hypotheses is presented as an easy-to-interpret odds ratio that represents the magnitude of the difference: 1–3 as weak, 3–10 as substantial, 10–30 as strong, 30–100 as very strong, and >100 as decisive [54].

## Results

Preliminary analysis of the dataset showed that a minimum number of the data were missing (all below 5%), so they were replaced with the mean value of the respective items.

**CFA and reliability.**   CFA results for a five-factor MWS solution displayed low fit [$\chi^2(N = 683, df = 220) = 972.17, p < .001$], RMSEA was adequate .07 [90%: .07– .08], TLI = .87 and CFI = .89 just below the .90 threshold. All items loaded significantly ($p < .001$) on appropriate factors, with adequate loadings' range: *Strong Will* [.46 – .70], *Skilful Will* [.47 – .60], *Good Will toward Self* [.36 – .72], *Good Will toward Others* [.47 – .79], and *Transpersonal Will* [.71 – .84]. Only IT10 ("*Doing something good for me makes me feel guilty.*") had a loading below .40. To shorten and strengthen the scale, IT10 was dropped from the scale, and reliability analysis suggested that also IT9 ("*I patiently look for the best way to do things.*"), IT22 ("*I recognize and use the inner qualities that can be useful to me in the pursuit of a goal.*"), and IT34 ("*I am willing to give up some of my time and projects for the sake of helping others.*") could be removed because did not contribute to the internal consistency of the scale.

Following the deletion of these items from the scale, analyses were repeated to confirm the factorial structure (CFA) and reliability. The CFA with 19 items confirmed the five-factor structure (see Fig 2) with an improved model fit ($\chi^2$ (142) = 507.63, $p < .001$; *TLI* = .91; *CFI* = .93; *RMSEA* = .06 [90%CI: .06–.07]). All items loaded significantly ($p < .001$) on appropriate factors and ranged from .54 to .83, in particular: *Strong Will* [.55 – .70], *Skillful Will* [.54 – .72], *Good Will toward Self* [.54 – .71], *Good Will toward Others* [.63 – .79], and *Transpersonal Will* [.73 – .83] (**Table 1**).

The correlations among five factors by and large replicated the results of Study 1. More precisely, the *Strong Will* correlated with *Good Will toward Self* ($r = .49$). The *Skillful Will* correlated with *Good Will toward Self* ($r = .46$), the *Good Will toward Others* ($r = .55$), and the *Transpersonal Will* ($r = .28$). Finally, the *Good Will toward Others* was related to the *Transpersonal Will* ($r = .33$). All the other relationships (range .05 – .17) were not significant.

The 19-item MWS internal consistency replicated previous results, all dimensions had fair to good Alpha and Omega coefficients, except *Skillful Will* (**Table 1**). The average inter-item correlation for Skillful Will was 36. The final 19-item scale is reported in Appendix B.

**Multigroup CFA.**   Multigroup CFA was conducted to test the measurement equivalence across genders and ages. The overall and comparative fit statistics of all four models (Model 0 – Model 3) are presented in Table 2. Results show an excellent fit of all tested models to the data, the differences in $\Delta CFI$ and $\Delta RMSEA$ values were lower than .01 indicating strict measurement invariance across genders and ages (**Table 2**).

**Validity.**   To assess the construct validity of the 19-item MWS, the correlations between its dimensions and other measures of related constructs were computed (**Table 3**).

**Table 1. Factor loadings, inter-item correlations, and internal consistency of the 19-item Multidimensional Will Scale (MWS).**

| Strong Will ($\alpha$ = .76 [CI: .73 - .79]; $\omega$ =. 77 [CI: .74 - .79]) | | | | |
| --- | --- | --- | --- | --- |
| Factor Loadings | | Item-total Correlations | Cronbach's $\alpha$ (if item dropped) | McDonald's $\omega$ (if item dropped) |
| IT17 | .55 | .51 | .74 | .74 |
| IT25 | .66 | .53 | .72 | .73 |
| IT30 | .61 | .60 | .70 | .70 |
| IT32 | .64 | .55 | .72 | .73 |
| IT33 | .70 | .51 | .73 | .74 |
| Skillful Will ($\alpha$ = .64 [CI: .59 - .69]; $\omega$ =. 66 [CI: .61 - .70]) | | | | |
| Factor Loadings | | Item-total Correlations | Cronbach's $\alpha$ (if item dropped) | McDonald's $\omega$ (if item dropped) |
| IT2 | .54 | .45 | .55 | – |
| IT35 | .72 | .53 | .44 | – |
| IT36 | .60 | .38 | .64 | – |
| Good Will toward Self ($\alpha$ = .73 [CI: .69 - .76]; $\omega$ =. 74 [CI: .70 - .77]) | | | | |
| Factor Loadings | | Item-total Correlations | Cronbach's $\alpha$ (if item dropped) | McDonald's $\omega$ (if item dropped) |
| IT3 | .65 | .51 | .67 | .70 |
| IT6 | .54 | .50 | .68 | .68 |
| IT14 | .71 | .61 | .61 | .61 |
| IT24 | .62 | .45 | .70 | .72 |
| Good Will toward Others ($\alpha$ = .80 [CI: .78 - .83]; $\omega$ =. 80 [CI: .78 - .83]) | | | | |
| Factor Loadings | | Item-total Correlations | Cronbach's $\alpha$ (if item dropped) | McDonald's $\omega$ (if item dropped) |
| IT5 | .68 | .59 | .77 | .77 |
| IT15 | .79 | .68 | .72 | .73 |
| IT19 | .63 | .55 | .79 | .79 |
| IT26 | .77 | .67 | .73 | .73 |
| Transpersonal Will ($\alpha$ = .83 [CI: .80 - .85]; $\omega$ =. 83 [CI: .81 - .85]) | | | | |
| Factor Loadings | | Item-total Correlations | Cronbach's $\alpha$ (if item dropped) | McDonald's $\omega$ (if item dropped) |
| IT4 | .73 | .68 | .77 | –[a] |
| IT13 | .80 | .67 | .77 | –[a] |
| IT23 | .83 | .71 | .74 | –[a] |

*Note.* [a] Omega if an item dropped requires at least 4 items to be computed.

For the *Strong Will*, medium to strong positive correlations were observed with the measures of Dispositional Optimism ($r$ = .45, $BF_{10}$>100), Sense of Coherence ($r$ = .42, $BF_{10}$>100), specifically for Manageability ($r$ = .58, $BF_{10}$>100), and Reflection ($r$ = 32, $BF_{10}$>100), Mastery ($r$ = .54, $BF_{10}$>100), Resilience ($r$ = .71, $BF_{10}$>100), Cheerfulness ($r$ = .35, $BF_{10}$>100), Seriousness ($r$ = .24, $BF_{10}$>100), Self-Control ($r$ = .40, $BF_{10}$>100), while a negative correlation was observed with Bad Mood ($r$ = -.46, $BF_{10}$>100).

The *Skillful Will* showed small to moderate positive correlations with Sense of Coherence ($r$ = .33, $BF_{10}$>100), Balance ($r$ = .20, $BF_{10}$>100), Reflection ($r$ = .38, $BF_{10}$>100), Resilience ($r$ = .18, $BF_{10}$>100), Seriousness ($r$ = .48, $BF_{10}$>100), Self-Control ($r$ = .36, $BF_{10}$>100).

The *Good Will toward Self* correlated positively with measures of Dispositional Optimism ($r$ = .35, $BF_{10}$>100), Sense of Coherence ($r$ = .27, $BF_{10}$>100), specifically for Manageability ($r$ = .30, $BF_{10}$>100), and Reflection ($r$ = .21, $BF_{10}$>100), Mastery ($r$ = .39, $BF_{10}$>100), Resilience

**Table 2. Fit statistics of the 19-item Multidimensional Will Scale (MWS) invariance models across genders and ages.**

| *Gender* | | | | | | | | |
|---|---|---|---|---|---|---|---|---|
| Model | $\chi^2$ (df) | CFI | RMSEA | Model comparison | $\Delta\chi2$ ($\Delta df$) | p | $\Delta CFI$ | $\Delta RMSEA$ |
| Configural | 571.24 (284) | .941 | .055 | - | - | - | - | - |
| Metric | 613.13 (298) | .936 | .056 | Metric-Configural | 41.89 (14) | < .005 | -.005 | .001 |
| Scalar | 642.73 (312) | .932 | .056 | Scalar–Metric | 29.6 (14) | < .005 | -.004 | .000 |
| Strict | 667.25 (331) | .931 | .055 | Strict–Scalar | 24.52 (19) | .25 | -.001 | -.001 |
| *Age* | | | | | | | | |
| Model | $\chi2$ (df) | CFI | RMSEA | Model comparison | $\Delta\chi2$ ($\Delta df$) | p | $\Delta CFI$ | $\Delta RMSEA$ |
| Configural | 607.326 (284) | .937 | .058 | - | - | - | - | - |
| Metric | 635.354 (298) | .934 | .058 | Metric-Configural | 28.028 (14) | < .01 | -.003 | .000 |
| Scalar | 703.415 (312) | .924 | .061 | Scalar–Metric | 68.061 (14) | < .005 | -.010 | .003 |
| Strict | 737.530 (331) | .920 | .060 | Strict–Scalar | 34.115 (19) | < .05 | -.004 | -.001 |

*Note*. df = degrees of freedom; CFI = comparative fit index; RMSEA = root mean square error of approximation; Δ = difference between nested models; p = probability value of Δχ2 test. Metric = equality of factor loadings; Scalar = Metric + equality of intercepts; Strict = Scalar + equality of error variances.

($r$ = .37, $BF_{10}$>100), Cheerfulness ($r$ = .38, $BF_{10}$>100), Seriousness ($r$ = .19, $BF_{10}$>100), Self-Control ($r$ = .25, $BF_{10}$>100), specifically for Self-Discipline ($r$ = .27, $BF_{10}$>100), while a moderate negative correlation was observed with Bad Mood ($r$ = -.36, $BF_{10}$>100).

**Table 3. Bivariate correlations between the 19-item Multidimensional Will Scale (MWS) and all the other measures of related constructs.**

| Variable | 1 | | 2 | | 3 | | 4 | | 5 | |
|---|---|---|---|---|---|---|---|---|---|---|
| 1. Strong Will | — | | | | | | | | | |
| 2. Skillful Will | .10 | | — | | | | | | | |
| 3. Good Will–Self | .36 | *** | .34 | *** | — | | | | | |
| 4. Good Will–Others | .02 | | .40 | *** | .07 | | — | | | |
| 5. Transpersonal Will | .11 | | .20 | *** | .11 | | .27 | *** | — | |
| 6. Dispositional Optimism | .45 | *** | .04 | | .35 | *** | .01 | | .21 | *** |
| 7. Sense of Coherence | .42 | *** | .33 | *** | .27 | *** | .23 | *** | .21 | *** |
| 8. Manageability | .58 | *** | .12 | | .30 | *** | .09 | | .16 | *** |
| 9. Balance | -.01 | | .20 | *** | .05 | | .13 | * | .09 | |
| 10. Reflection | .32 | *** | .38 | *** | .21 | *** | .28 | *** | .20 | *** |
| 11. Mastery | .54 | *** | .06 | | .39 | *** | -.05 | | .01 | |
| 12. Resilience | .71 | *** | .18 | *** | .37 | *** | .13 | * | .24 | *** |
| 13. Cheerfulness | .35 | *** | .11 | | .38 | *** | .20 | *** | .20 | *** |
| 14. Seriousness | .24 | *** | .48 | *** | .19 | *** | .33 | *** | .27 | *** |
| 15. Bad Mood | -.46 | *** | 0.10 | | -.36 | *** | -.07 | | -.09 | |
| 16. Self Control | .40 | *** | .36 | *** | .25 | *** | .22 | *** | .20 | *** |
| 17. Impulse Control | .28 | *** | .29 | *** | .16 | *** | .19 | *** | .15 | *** |
| 18. Self-discipline | .40 | *** | .33 | *** | .27 | *** | .18 | *** | .19 | *** |

*Note*. N = 683

*$BF_{10}$>10

**$BF_{10}$>30

***$BF_{10}$>100.

The *Good Will toward Others* showed weak to moderate positive correlations with measures of Sense of Coherence ($r = .23$, $BF_{10} > 100$), Reflection ($r = .28$, $BF_{10} > 100$), Cheerfulness ($r = .20$, $BF_{10} > 100$), Seriousness ($r = .33$, $BF_{10} > 100$), Self-Control ($r = .22$, $BF_{10} > 100$), Balance ($r = .13$, $BF_{10} > 10$), and Resilience ($r = .13$, $BF_{10} > 10$).

Finally, with weak positive correlations between the *Transpersonal Will* and measures of Dispositional Optimism ($r = .21$, $BF_{10} > 100$), Sense of Coherence ($r = .21$, $BF_{10} > 100$), Manageability ($r = .16$, $BF_{10} > 100$), Reflection ($r = .20$, $BF_{10} > 100$), Resilience ($r = .24$, $BF_{10} > 100$), Cheerfulness ($r = .20$, $BF_{10} > 100$), Seriousness ($r = .27$, $BF_{10} > 100$), and Self-Control ($r = .20$, $BF_{10} > 100$) were found.

## Discussion

Step 3 allowed the further refinement of the MWS toward its final version. The five-factor model of the 19-item scale showed an adequate fit with good factor loadings. This result confirmed that, according to Assagioli's conceptualization [5], the will is a multidimensional construct with some facets more closely related to each other, such as *Strong Will* and *Good Will toward Self*, and *Good Will toward Others* and *Transpersonal Will*. The internal consistency confirmed the trend that was observed in the previous analyses, with a slight improvement for two dimensions (*Good Will toward Self* and *Good Will toward Others*). The reliability was just below the threshold for *Skillful Will* probably due to the fact that it consists of only three items quite heterogeneous in content. Analyses of measurement invariance across genders and ages confirmed the measurement equivalence of the MWS across groups. Therefore, the scale can be employed with younger and older respondents of both genders to make unbiased comparisons [55].

Although the current validity investigation had an exploratory purpose, the observed relationships provide some evidence for a correct operationalization of the five dimensions of the will construct. Indeed, subscales correlated with the measures of self-control and the sense of coherence scale. The former is related to the ability to resist temptations and modify dominant responses through self-regulation mechanisms [42], and the latter regards the understanding that life comprises positive and negative experiences and that achieving stability necessitates integrating different cognitive skills such as wisdom, meaning making, and resiliency [36]. The correlations with dispositional optimism were found for *Strong*, *Transpersonal*, and *Good Will toward Self*. Defined as a personal disposition to attend generally positive outcomes in life [56], and positively related to psychological well-being, physical health, and positive health behaviors, this relationship reinforces the definition of will as an aspect linked to self-fulfilling goals and personal well-being.

## Conclusion

The aim of this study was the development of a scale designed to assess different aspects of the will based on the theoretical model of Assagioli, an Italian psychiatrist whose ideas are largely applied to psychoanalysis and counseling treatments. Although giving an operative definition of this theoretical model was challenging, a quantitative measure of a multidimensional construct that consists of the different aspects of will (Strong, Skillful, Good toward Self, Good toward Others, and Transpersonal) was obtained. Throughout several phases, a scale with good psychometric properties was obtained, the five-factor structure derived from the theoretical model was confirmed, and validity evidence was provided.

The MWS could be used to understand the possible benefits of assessing the will in connection to individual and relational well-being or, vice versa, the impact of a poor will on distress. Additionally, it may represent a useful tool for initial screening, for example, in patients who

are required to follow rehabilitation and health maintenance programs. The will could play an important role in determining whether the patient will consistently follow the proposed treatment and, therefore, measuring the will can offer insights to develop specific interventions for those who lack it.

The current work is not without limitations. To overcome the sample bias of the first study (Step 2), which was conducted on a sample of university students, in the second study (Step 3) the "snowball" sampling method was employed. However, this sample might not be representative of the Italian general population. Future studies should attempt to replicate these findings by including broader and more heterogeneous samples. At the same time, the scale should be tested on clinical samples (e.g., patients with psychiatric or psychosomatic pathologies) to prove the suitability and utility of the MWS in this domain. Specifically, since volition can be conceptualized as a salutogenic construct, future studies using the MWS will lead to a deeper understanding of the relationships with well-being and the prevention of physical and psychological pathologies, preferably through prospective studies. Furthermore, validity evidence was provided in the current study, but extending the scale validity by including measures developed to assess volition and motivation would be desirable. Finally, since the scale was developed in Italian, it would be worth translating and validating it in other languages, to confirm its psychometric properties, and to make it available for use in different linguistic contexts.

In conclusion, whereas the current findings need to be confirmed and extended, the MWS appears to be an efficient multidimensional measure that allows researchers, therapists, and counselors to assess Assagioli's construct of will.

## Supporting information

**S1 Table. A. Descriptive statistics for 38-item Multidimensional Will Scale (MWS). B.** Factor Loadings, inter-item correlations and internal consistency of the 30-item solution of the Multidimensional Will Scale (MWS). **C.** Factor Loadings, inter-item correlations, and internal consistency of the 23-item Multidimensional Will Scale (MWS).
(DOCX)

**S1 Appendix. The initial 38 items of the Multidimensional Will Scale (MWS).**
(DOCX)

**S2 Appendix. The final 19 items of the Multidimensional Will Scale (MWS).**
(DOCX)

**S1 File.**
(SAV)

**S2 File.**
(SAV)

**S3 File.**
(SAV)

## Acknowledgments

The authors thank the trainers, psychotherapists, psychologists, and volunteers of the Italian Psychosynthesis Institute for their kind support of the research and particularly are grateful to: Piero Ferrucci, Nives Favero, Vittorio Viglienghi, Silvia Messina, Eleonora Fazzini, Serena Stanghellini, Erika Agresti, Petra Guggisberg Nocelli, Piero Marovelli, Alessandro Toccafondi,

Alberto Nannicini, Linda Cecconi, Elena Morbidelli, Giuditta Greco, Luigi Padovese, Carla Pellegrini.

## Author Contributions

**Conceptualization:** Andrea Bonacchi.

**Data curation:** Georgia Marunic, Carlotta Tagliaferro, Rebecca Boschi, Francesca Chiesi.

**Formal analysis:** Georgia Marunic, Carlotta Tagliaferro, Chloe Lau, Francesca Chiesi.

**Investigation:** Andrea Bonacchi, Francesca Chiesi.

**Methodology:** Andrea Bonacchi, Chloe Lau, Francesca Chiesi.

**Project administration:** Andrea Bonacchi.

**Supervision:** Francesca Chiesi.

**Writing – original draft:** Andrea Bonacchi, Georgia Marunic, Carlotta Tagliaferro, Chloe Lau, Francesca Chiesi.

**Writing – review & editing:** Georgia Marunic, Carlotta Tagliaferro, Chloe Lau, Francesca Chiesi.

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
