## [Decision Letter · Decision Letter 0]

31 Jul 2023

PONE-D-23-06766Assessment of will from psychodynamic theory: The development of the Multidimensional Will Scale

PLOS ONE

Dear Dr. Bonacchi,

Thank you for submitting your manuscript to PLOS ONE. After careful consideration, we feel that it has merit but does not fully meet PLOS ONE’s publication criteria as it currently stands. 

Therefore, we invite you to submit a revised version of the manuscript that addresses the points raised during the review process.

Your work is truly valuable for the mental health community; reading it made me believe it should be considered. However, there are some significant issues that our exceptional specialist reviewer has pointed out regarding the logical flow and reliability of the psychometric data, which in this instance are of particular concern. 

Please thoroughly review them and make the necessary adjustments and/or suitable clarifications.

After your submission of the revised manuscript,a second round of blind-review process will be run. 

Please submit your revised manuscript by Sep 14 2023 11:59PM

If you will need more time than this to complete your revisions, please reply to this message or contact the journal office at plosone@plos.org. Please include the following items when submitting your revised manuscript:

We look forward to receiving your revised manuscript.

Kind regards,

Silva Ibrahimi, PhD

Academic Editor

PLOS ONE

Journal Requirements:

2. Please provide additional details regarding ethical approval in the body of your manuscript. In the Methods section, please ensure that you have specified the name of the IRB/ethics committee that approved your study.

Reviewers' comments:

Reviewer's Responses to Questions

**Comments to the Author**

1. Is the manuscript technically sound, and do the data support the conclusions?

Reviewer #1: Partly

2. Has the statistical analysis been performed appropriately and rigorously? 

Reviewer #1: No

3. Have the authors made all data underlying the findings in their manuscript fully available?

Reviewer #1: Yes

4. Is the manuscript presented in an intelligible fashion and written in standard English?

Reviewer #1: No

5. Review Comments to the Author

Reviewer #1: Thank you for giving me this opportunity to review the manuscript (No: PONE-D-23-06766).

This study aimed to develop and validate a Multidimensional Will Scale (MWS). I applaud the authors’ efforts to identify the dimensions of will according to Assagioli’s model and develop the preliminary 38-item version of the MWS in three phases, examine the factor structure of the 38-item MWS using exploratory factor analysis (Study 1), and identify the best-fit model using confirmatory factor analysis (Study 2).

Major issues:

1. The item selection procedure and the cut-off values used to remove items are not indicated in lines 258 to 260. As a result, the decision to remove items 7, 16, and 27 because of the skewness and/or kurtosis values does not seem justifiable. The values were below 2.00 and are considered acceptable to some researchers. The authors are urged to explain the reason for using skewness and kurtosis, rather than factor loading, to determine the items that shall be removed. Finally, the kurtosis value of item 27 reported in line 276 is different from the value reported in S1 Table.

2. Lines 287 to 286, the number of factors to retain in the parallel analysis is not decided by the eigenvalue > 1 criterion, but by the number of eigenvalues (generated from the dataset) that are larger than the corresponding random eigenvalues.

3. Lines 291 to 292, “the analyses for a 31-item solution, with the number of factors fixed at five, were repeated.” The writing does not indicate whether the researchers delete one of the four items (i.e., items 18, 21, 29, 31) each time and then rerun the EFA. In addition, it is not justifiable to fix the number of factors. Note that the number of factors is likely to change when an item is removed.

4. Line 293, “IT12 (r=.31) has been eliminated as statistically non-significant.” It is not clear if the reported r-value refers to factor loading. Moreover, the authors did not mention that statistical significance is one of the criteria for determining item removal.

5. Lines 301 to 302, “The content analysis led to moving these two items from the Skillful to the Strong Will dimension” The writing seems to indicate that the authors think that it is acceptable to keep items 22 and 32 to Strong Will dimension. The items, however, were evaluated by experts and recommended to be loaded on Skillful Will dimension. The inconsistent decisions require further explanation.

6. Lines 319 to 320 and Lines 342 to 343, items 9 and 10 with a factor loading below 0.40 were not removed to maintain the Cronbach alpha. This practice, again, does not seem justifiable.

7. Lines 329, Is the result of the scree test consistent with the parallel analysis result? It is confusing why the authors referred to the scree test result.

8. Lines 506 to 513, items 10, 9, 22, and 34 were removed and the retained items were submitted to CFA. The CFA results shall be interpreted as exploratory rather than confirmatory. A new set of data shall be collected to confirm the revised structure/model.

Minor issue

1. The authors use the term will and volition in the literature review. It is important to clarify if the two terminologies are conceptually equivalent.

2. Lines 246 to 249, explain the reason to run two EFAs using PC and FA respectively.

I did not review the Discussion because the abovementioned major issues lead me to doubt the appropriateness of the selected items.

6. PLOS authors have the option to publish the peer review history of their article (what does this mean?). If published, this will include your full peer review and any attached files.

Reviewer #1: No

---

## [Author Response · Author response to Decision Letter 0]

8 Sep 2023

Dear Reviewer, 

Thank you for the opportunity to improve our manuscript. We have addressed all the issues raised, which can be found in the uploaded file "Response to Reviewers".

Kind regards,

The Authors

---

## [Decision Letter · Decision Letter 1]

12 Dec 2023

PONE-D-23-06766R1Assessment of will from psychodynamic theory: The development of the Multidimensional Will ScalePLOS ONE

Dear Dr. Bonacchi,

Thank you for submitting your manuscript to PLOS ONE. After careful consideration, we feel that it has merit but does not fully meet PLOS ONE’s publication criteria as it currently stands. Therefore, we invite you to submit a revised version of the manuscript that addresses the points raised during the review process. Based on our expert reviewers' comments and opinions the manuscript still need a throughout revision,especially in terms of a pure  psychodynamic background . Please also be sure to address the first reviewer concerns  as he highlighted. Please submit your revised manuscript by Jan 26 2024 11:59PM. If you will need more time than this to complete your revisions, please reply to this message or contact the journal office at plosone@plos.org. Please include the following items when submitting your revised manuscript:A rebuttal letter that responds to each point raised by the academic editor and reviewer(s). You should upload this letter as a separate file labeled 'Response to Reviewers'.A marked-up copy of your manuscript that highlights changes made to the original version. You should upload this as a separate file labeled 'Revised Manuscript with Track Changes'.An unmarked version of your revised paper without tracked changes. You should upload this as a separate file labeled 'Manuscript'.

We look forward to receiving your revised manuscript.

Kind regards,

Silva Ibrahimi, PhD

Academic Editor

PLOS ONE

Reviewers' comments:

Reviewer's Responses to Questions

**Comments to the Author**

1. If the authors have adequately addressed your comments raised in a previous round of review and you feel that this manuscript is now acceptable for publication, you may indicate that here to bypass the “Comments to the Author” section, enter your conflict of interest statement in the “Confidential to Editor” section, and submit your "Accept" recommendation.

Reviewer #2: (No Response)

Reviewer #3: (No Response)

2. Is the manuscript technically sound, and do the data support the conclusions?

Reviewer #2: Partly

Reviewer #3: Partly

3. Has the statistical analysis been performed appropriately and rigorously? 

Reviewer #2: No

Reviewer #3: Yes

4. Have the authors made all data underlying the findings in their manuscript fully available?

Reviewer #2: Yes

Reviewer #3: Yes

5. Is the manuscript presented in an intelligible fashion and written in standard English?

Reviewer #2: Yes

Reviewer #3: No

6. Review Comments to the Author

Reviewer #2: ID: PONE-D-23-06766

Title: Assessment of will from psychodynamic theory: The development of the Multidimensional Will Scale

Thank you for providing a chance to review this manuscript.

Abstract

Overall: The Abstract is confusing, it is recommended to add subheadings in the order of “Background”, “Objective”, “Methods”, “Results”, and “Conclusion”, and to check whether each section is clearly explained.

Line 34, Page 2: “N=587; Mage=21.55, SDage=4.14, and N=683; Mage = 34.09, SDage = 16.27, respectively”, It is recommended to move to “Study 1” and “Study 2” respectively.

Line 34-42, Page 2: In addition to the results of the various analyses, the results of important statistical values are also those that should be presented in the abstract and the authors are invited to add them.

Introduction

Line 51-56, Page 3: What is the key meaning that needs to be conveyed in this paragraph? Why the sudden reference to “voluntary acts and behaviors enables scholars” when the importance and definition of will is being described? The author's presentation lacks clarity and focus on the topic.

Line 57-73＆Line 74-81, Page 2-3: The author devotes a tremendous amount of space to description in these two paragraphs, but still does not provide a clear answer to the question posed at the end of the previous paragraph, “Over time, numerous attempts have been made to define and discern between different conceptualizations of the will, willpower, and human volition”. At the same time, I don't think such a lengthy history of development is particularly relevant to your research background, and these two paragraphs need to be merged and greatly abbreviated.

Line 82-92, Page 4: The author’s research concepts seem to keep jumping around; what is the relationship between “volition” and “will”?

Line 98, Page 4: “That need to include the development of assessment tools”, your research focuses on tool development, please describe in detail. I don’t understand from your statement why there is a need to assess the development of tools?

Theoretical background: Assagioli’s model of will

Overall: Overall, the description of the theoretical framework could have been interspersed with the “Introduction”, and I do not see the need for it to be placed under a separate subtitle, which would be detrimental to the coherence of the contextualization.

Overall: Same problem, the description is too long, so detailed that the author can't catch the point, I suggest the author to make cuts.

Designing the Multidimensional Will Scale (MWS)

Overall: It is suggested that the presentation of the subtitle be modified by adding the specific steps of the research undertaken after “Phase I”, etc.

Phase I, Page 7-8: “In the first stage, the purpose was to adequately operationalize the construct to ensure face and content validity”, the description of the entire paragraph does not mention the tests and results of surface sanitization and content validity, which are necessary to be accounted for.

Overall: The authors describe multiple stages of modifying and validating the content of the scale, but how was the original pool of entries for the scale determined? The authors do not have a clear statement in this area.

Study 1: Testing and revising the Multidimensional Will Scale (MWS)

Participants and sample size, Page 9: The representation of the demographic characteristics of the sample is necessary. Authors were asked to add tables showing relevant data to demonstrate that the sample collected was representative.

Line 218-221, Page 9-10: “The sample size was established referring to the requirements to develop a scale through factor analysis that depends on the number and type of the items, the extent of correlations between items in the population, average commonality, the number of factors, and the stability of the factor structure”, what specific requirements have been complied with? How was the sample size synthesized? The authors should clearly list this.

Line 225-227, Page 10: “Approximately 66% of participants identified as cisgender women, 33% as cisgender men, and 1% preferred not to say. The majority were students (91%) from a large university in central Italy”, why is there such a large distribution of females and students, given that this is a survey facing an adult population? I question the representativeness of the sample. As well, the authors' sampling method was?

Line 236-237, Page 10: “A subsample of participants (N = 228; 39% of the original sample) completed the MWS again in a four-to-five-week interval”, what is the basis for determining the time interval?

Line 239-243, Page 10: “Otherwise, the arithmetic mean of each item replaced the missing data”, is there literature to support your choice of mean values to supplement missing values?

Item descriptives, Page 12-13: Is there a theory to support the decision to simply delete items when there is a deviation from normality? The test for normality may vary with the sample taken, and I don't think it's reasonable to handle it that way.

Overall: I do not believe that a discussion after only a preliminary analysis is necessary and suggest that the authors merge the discussion of the two parts of the study.

Study 2: Dimensionality, gender and age invariance, and validity of the Multidimensional Will Scale (MWS)Longitudinal invariance analyses of the Chinese EPDS

Measures, Page 18-20: Please list in detail the process of using the translation of the scale with validation in the local language.

Line 510-512, Page 21: What is the reason for choosing the change in CFI and RMSEA as the cutoff value for determining invariance?4

Line 564-565, Page 24＆Table 2, Page 25: “Results show an excellent fit of all tested models to the data, the differences in ΔCFI and ΔRMSEA values were lower than .01 and .015, respectively”, please check the data again, isn't the ΔCFI for scalar invariance in the age subgroup outside the recommended range?

Discussion, Page 28-30: The discussion is more of a restatement of the results, and the authors are advised to move to a deeper level of discussion.

Conclusion

Overall: The conclusion is not concise, it uses too much space and contains too much content, and the author is advised to rewrite it, preferably in two to three sentences.

This study is innovative and fills the research gap in this field. However, it still has certain shortcomings. The article structure of this paper has a big problem, which is confusing and unsystematic, and needs a big change; and the author's account is more confusing, unable to catch the key points, and using too much space, which needs to be partially reduced. At the same time, there are certain problems at the research design and data analysis. I am sorry to give the decision of rejection and look forward to seeing the progress of this manuscript.

Thank you and my best,

Your reviewer

Reviewer #3: The authors need to more fully explain the theoretical model on which they are basing their study, and the relevance of the research as designed and conducted. It is also important to correct the presumption that this an empirical study of psychodynamic psychotherapy--psychosynthesis holds very little concordance with psychoanalysis/psychodynamic psychotherapy even though its founder studied psychoanalysis.

7. PLOS authors have the option to publish the peer review history of their article (what does this mean?). If published, this will include your full peer review and any attached files.

Reviewer #2: No

Reviewer #3: No

---

## [Author Response · Author response to Decision Letter 1]

13 Feb 2024

Dear Editor,

Thank you for your letter dated December 12, 2023. We are sincerely grateful that you are willing to consider a revised draft of our manuscript for potential publication in the PLOS One.

We thank you and the Reviewers for their insightful comments and suggestions. We have revised the manuscript following their feedback. Attached, you will find a copy of the revised manuscript, which includes tracked changes, along with an unmarked version. Additionally, we have provided detailed point-by-point responses to the reviewer's comments. In the revised manuscript, all modifications made to the previous version are tracked. 

We would also like to take this opportunity to thank you for helping us to improve the quality of our manuscript. Nonetheless, we’d like to remember that we sent the paper on March 13, 2023. After four months (July 31, 2023), we received a detailed review (Review #1) along with some editor’s standard indications (style requirement, ethics statement). We sent the revised version on September 8th, 2023. After three months (December 12, 2023), we received two different new reviews by Reviewer #2 and Reviewer #3. Although we agree that a publication must follow rigorous evaluations, we believe that a less long and complicated procedure might have shortened the process. Thus, at this stage, we hope to receive an answer as soon as possible.

Kind regards,

Andrea Bonacchi

Reviewer #2:

Abstract

Overall: The Abstract is confusing, it is recommended to add subheadings in the order of “Background”, “Objective”, “Methods”, “Results”, and “Conclusion”, and to check whether each section is clearly explained.

Line 34, Page 2: “N=587; Mage=21.55, SDage=4.14, and N=683; Mage = 34.09, SDage = 16.27, respectively”, It is recommended to move to “Study 1” and “Study 2” respectively. Line 34-42, Page 2: In addition to the results of the various analyses, the results of important statistical values are also those that should be presented in the abstract and the authors are invited to add them.

Response: The abstract was changed following these suggestions.

Introduction

Line 51-56, Page 3: What is the key meaning that needs to be conveyed in this paragraph? Why the sudden reference to “voluntary acts and behaviors enables scholars” when the importance and definition of will is being described? The author's presentation lacks clarity and focus on the topic.

Line 57-73＆Line 74-81, Page 2-3: The author devotes a tremendous amount of space to description in these two paragraphs, but still does not provide a clear answer to the question posed at the end of the previous paragraph, “Over time, numerous attempts have been made to define and discern between different conceptualizations of the will, willpower, and human volition”. At the same time, I don't think such a lengthy history of development is particularly relevant to your research background, and these two paragraphs need to be merged and greatly abbreviated.

Line 82-92, Page 4: The author’s research concepts seem to keep jumping around; what is the relationship between “volition” and “will”? Line 98, Page 4: “That need to include the development of assessment tools”, your research focuses on tool development, please describe in detail. I don’t understand from your statement why there is a need to assess the development of tools? 

Theoretical background: Assagioli’s model of will

Overall: Overall, the description of the theoretical framework could have been interspersed with the “Introduction”, and I do not see the need for it to be placed under a separate subtitle, which would be detrimental to the coherence of the contextualization.

Overall: Same problem, the description is too long, so detailed that the author can't catch the point, I suggest the author to make cuts.

Response: The introduction was shortened and changed following these suggestions. 

Designing the Multidimensional Will Scale (MWS)

Overall: It is suggested that the presentation of the subtitle be modified by adding the specific steps of the research undertaken after “Phase I”, etc.

Response: Headings and sub-headings were changed following these suggestions and to give the paper a clearer structure. 

Phase I, Page 7-8: “In the first stage, the purpose was to adequately operationalize the construct to ensure face and content validity”, the description of the entire paragraph does not mention the tests and results of surface sanitization and content validity, which are necessary to be accounted for. 

Overall: The authors describe multiple stages of modifying and validating the content of the scale, but how was the original pool of entries for the scale determined? The authors do not have a clear statement in this area. I

The paper Response: In “Operational definition and Item formulation” section we explained how the initial pool of items was obtained and in the “Content Validity” section we detailed the results of content analysis. 

Study 1: Testing and revising the Multidimensional Will Scale (MWS)

Participants and sample size, Page 9: The representation of the demographic characteristics of the sample is necessary. Authors were asked to add tables showing relevant data to demonstrate that the sample collected was representative.

Response: The first sample was not representative because it was a convenience sample. The following study (called now Step 3) was conducted with a different sample to amend this flaw. Moreover, we acknowledged this limitation of the study in the Constraints on Generality section.

Line 218-221, Page 9-10: “The sample size was established referring to the requirements to develop a scale through factor analysis that depends on the number and type of the items, the extent of correlations between items in the population, average commonality, the number of factors, and the stability of the factor structure”, what specific requirements have been complied with? How was the sample size synthesized? 

Response: In the Participants section of Step 2 we better explained the choice of the sample size.

Line 225-227, Page 10: “Approximately 66% of participants identified as cisgender women, 33% as cisgender men, and 1% preferred not to say. The majority were students (91%) from a large university in central Italy”, why is there such a large distribution of females and students, given that this is a survey facing an adult population? I question the representativeness of the sample. As well, the authors' sampling method was?

Response: See above. 

Line 236-237, Page 10: “A subsample of participants (N = 228; 39% of the original sample) completed the MWS again in a four-to-five-week interval”, what is the basis for determining the time interval? 

Response: This interval is recommended in the literature (e.g., DeVellis, 2012; Polit, 2014) especially when the measure refers to trait-like attribute. A trait has high stability; thus no significant changes are expected. We detailed this point in the Measure and Procedure section of Step 2.

Line 239-243, Page 10: “Otherwise, the arithmetic mean of each item replaced the missing data”, is there literature to support your choice of mean values to supplement missing values?

Response: This is one of the ways to deal with missing data (Little and Rubin, 2002). We are aware that this method can limit data variability. However, in this case missing data were very few (never exceeded 1% of the responses, i.e., max 5/6 out of 587). Therefore, we can be confident that the missing data imputation method did not impact on the analyses.

Item descriptives, Page 12-13: Is there a theory to support the decision to simply delete items when there is a deviation from normality? The test for normality may vary with the sample taken, and I don't think it's reasonable to handle it that way.

Response: Deviation from normality means that the variability in the responses is in someway reduced (responses tend to concentrate to some response options and, as a result, we observe skewed and/or kurtotic distribution). Since we were in a phase of development of the scale, we preferred to exclude items that showed these characteristics. 

Overall: I do not believe that a discussion after only a preliminary analysis is necessary and suggest that the authors merge the discussion of the two parts of the study.

Response: We thank you for this suggestion, but we deem it necessary at this point of the paper to briefly sum up the obtained results before introducing Step 3 of the scale development. Thus, the Discussion was maintained but drastically shortened.

Study 2: Dimensionality, gender and age invariance, and validity of the Multidimensional Will Scale (MWS).Line 510-512, Page 21: What is the reason for choosing the change in CFI and RMSEA as the cutoff value for determining invariance?

Response: As mentioned in the Analysis section of Step 3, we followed the literature (Chen (2007) http://dx.doi.org/10.1080/10705510701301834; Cheung & Rensvold (2002) http://10.1207/s15328007sem0902_5); Rutkowski & Svetina, (2014). https://doi.org/10.1177/0013164413498257

Line 564-565, Page 24＆Table 2, Page 25: “Results show an excellent fit of all tested models to the data, the differences in ΔCFI and ΔRMSEA values were lower than .01 and .015, respectively”, please check the data again, isn't the ΔCFI for scalar invariance in the age subgroup outside the recommended range? 

Response: The value was .010. The error was due to rounding decimals. We corrected the values in Table 2. 

Discussion, Page 28-30: The discussion is more of a restatement of the results, and the authors are advised to move to a deeper level of discussion.

Conclusion Overall: The conclusion is not concise, it uses too much space and contains too much content, and the author is advised to rewrite it, preferably in two to three sentences.

Response: The Discussion and the Conclusion were changed following these suggestions. 

Reviewer #3 

Comment: The authors need to more fully explain the theoretical model on which they are basing their study, and the relevance of the research as designed and conducted. It is also important to correct the presumption that this an empirical study of psychodynamic psychotherapy--psychosynthesis holds very little concordance with psychoanalysis/psychodynamic psychotherapy even though its founder studied psychoanalysis.

Response: Thanks for this suggestion. We modified the Introduction (and the Title) accordingly.

---

## [Decision Letter · Decision Letter 2]

19 Apr 2024

PONE-D-23-06766R2The development of the Multidimensional Will ScalePLOS ONE

Dear Dr. Bonacchi,

Thank you for submitting your manuscript to PLOS ONE. After careful consideration, we feel that it has merit but does not fully meet PLOS ONE’s publication criteria as it currently stands. Therefore, we invite you to submit a revised version of the manuscript that addresses the points raised during the review process.

We look forward to receiving your revised manuscript.

Kind regards,

Yansong Li

Academic Editor

PLOS ONE

Journal Requirements:

Reviewers' comments:

Reviewer's Responses to Questions

**Comments to the Author**

1. If the authors have adequately addressed your comments raised in a previous round of review and you feel that this manuscript is now acceptable for publication, you may indicate that here to bypass the “Comments to the Author” section, enter your conflict of interest statement in the “Confidential to Editor” section, and submit your "Accept" recommendation.

Reviewer #2: (No Response)

Reviewer #4: (No Response)

2. Is the manuscript technically sound, and do the data support the conclusions?

Reviewer #2: Yes

Reviewer #4: Yes

3. Has the statistical analysis been performed appropriately and rigorously? 

Reviewer #2: Yes

Reviewer #4: Yes

4. Have the authors made all data underlying the findings in their manuscript fully available?

Reviewer #2: Yes

Reviewer #4: Yes

5. Is the manuscript presented in an intelligible fashion and written in standard English?

Reviewer #2: Yes

Reviewer #4: Yes

6. Review Comments to the Author

Reviewer #2: ID: PONE-D-23-06766R2

Title: The development of the Multidimensional Will Scale

Thank you for providing a chance to review this manuscript.

Recommendation: Minor revise.

The author has made careful revisions and responses, and the quality of the article has improved considerably. Congratulations! I have only the following minor issues to express my doubts:

Detailed information:

Title

Overall: Personally, I think the overly short title does not reflect the specificities of your study.

Abstract

Overall: Formatting issues need more attention, such as spaces before and after “=”, italicizing “N”, etc.

Overall: “(TLI = .91; CFI = .93; RMSEA = .06 [90%CI: .06‒.07])”, note the explanation of abbreviations when they are used for the first time.

Step 2: Results:

Overall: Why don't you use charts and graphs to show the results of the EFA more clearly, instead of using complex textual representations?

Step 3: Method

Overall: Why is it not consistent here with the previous text, with subheadings separating the analyzed sections.

Step 3: Discussion

Overall: An account of the research highlights, limitations, and future research directions related to this study is lacking and should be added.

Thank you and my best,

Your reviewer

Reviewer #4: Abstract:

• The sample needs labeling, such as Sample 1, Sample 2, to clarify whether there is any overlap between samples. Subsamples should also be coded for clarity, as the current description can lead to confusion.

• In the abstract, Step 2 needs a clearer description detailing the number of factors identified and their respective names.

• The presentation of the CFI model parameters is incomplete.

• The reliability section should specify the exact values for clarity and precision.

Introduction:

• The introduction contains an excessive focus on philosophical concepts and history, with a noticeable lack of recent psychological research on will.

• There is an absence of mention of other will-related questionnaires, multidimensional scale and psychological variables related to will (e.g., variables from questionnaires used as benchmarks). Recommend papers:

Reise, S. P., Bonifay, W. E., & Haviland, M. G. (2013). Scoring and Modeling Psychological Measures in the Presence of Multidimensionality. Journal of Personality Assessment, 95(2), 129–140. https://doi.org/10.1080/00223891.2012.725437

Osman, A., Lamis, D. A., Freedenthal, S., Gutierrez, P. M., & McNaughton-Cassill, M. (2014). The Multidimensional Scale of Perceived Social Support: Analyses of Internal Reliability, Measurement Invariance, and Correlates Across Gender. Journal of Personality Assessment, 96(1), 103–112. https://doi.org/10.1080/00223891.2013.838170

Wang, Y., Orosz, G., Chen, X., Miao, C., & Li, Y. (2024). Psychometric evaluation of the Chinese version of the Multidimensional Competitive Orientation Inventory. Scientific Reports, 14(1), 6591.

Step 1:

• A critical ratio analysis is necessary to demonstrate the item discriminability for each question.

Step 2:

• The sample exhibits a gender imbalance that needs addressing.

• The reported degrees of freedom (df) value for Bartlett’s test of sphericity appears unusual and should be verified.

Step 3:

• The rationale for item deletion needs to be more comprehensive to justify the decisions made during this step.

7. PLOS authors have the option to publish the peer review history of their article (what does this mean?). If published, this will include your full peer review and any attached files.

Reviewer #2: No

Reviewer #4: No

---

## [Author Response · Author response to Decision Letter 2]

27 May 2024

Dear Editor,

Thank you for your letter dated April 23, 2024. We are sincerely grateful that you are willing to consider a revised draft of our manuscript for potential publication in PLOS One.

We thank you and the Reviewers for their additional comments and suggestions. We have revised the manuscript following their feedback. Attached, you will find a copy of the revised manuscript, which includes tracked changes, along with an unmarked version. Moreover, we have provided detailed point-by-point responses to the reviewer's comments.

Kind regards,

Andrea Bonacchi

.

Reviewer #2:

Overall: Personally, I think the overly short title does not reflect the specificities of your study

Response: Thank you for your suggestion. We propose a title reflecting more our study: “Measuring Strong, Skillful, Good and Transpersonal Will: The development of the Multidimensional Will Scale”. 

Abstract

Formatting issues need more attention, such as spaces before and after “=”, italicizing “N”, etc.

Response: Thank you for noticing that. We modified accordingly.

Overall: “(TLI = .91; CFI = .93; RMSEA = .06 [90%CI: .06‒.07])”, note the explanation of abbreviations when they are used for the first time. 

Response: Thak you, but, in line with many other published papers, we prefer not to include the explanation of fit indices abbreviations in the Abstract.

Step 2: Results:

Why don't you use charts and graphs to show the results of the EFA more clearly, instead of using complex textual representations?

Response: We apologize if they were missing in the last version, but figures are included in the manuscript submission and mentioned in the text.

Step 3: Method: 

Why is it not consistent here with the previous text, with subheadings separating the analyzed sections.

Response: We employed different subheadings because we performed partially different analyses (e.g., CFA instead of EFA)

Discussion 

An account of the research highlights, limitations, and future research directions related to this study is lacking and should be added.

Response: We added e description of some limitations and related research directions in the Discussion section,

Reviewer #4: 

Abstract:

The sample needs labeling, such as Sample 1, Sample 2, to clarify whether there is any overlap between samples. Subsamples should also be coded for clarity, as the current description can lead to confusion. 

In the abstract, Step 2 needs a clearer description detailing the number of factors identified and their respective names.

The presentation of the CFI model parameters is incomplete.

The reliability section should specify the exact values for clarity and precision.

Response: All the requested changes were made following the suggestions.

Introduction:

The introduction contains an excessive focus on philosophical concepts and history, with a noticeable lack of recent psychological research on will.

There is an absence of mention of other will-related questionnaires, multidimensional scale and psychological variables related to will (e.g., variables from questionnaires used as benchmarks)

Recommend papers: Reise, S. P., Bonifay, W. E., & Haviland, M. G. (2013). Scoring and Modeling Psychological Measures in the Presence of Multidimensionality. Journal of Personality Assessment, 95(2), 129–140.https://doi.org/10.1080/00223891.2012.725437

Osman, A., Lamis, D. A., Freedenthal, S., Gutierrez, P. M., & McNaughton-Cassill, M. (2014). The Multidimensional Scale of Perceived Social Support: Analyses of Internal Reliability, Measurement Invariance, and Correlates Across Gender. Journal of Personality Assessment, 96(1), 103–112.https://doi.org/10.1080/00223891.2013.838170

Wang, Y., Orosz, G., Chen, X., Miao, C., & Li, Y. (2024). Psychometric evaluation of the Chinese version of the Multidimensional Competitive Orientation Inventory. Scientific Reports, 14(1), 6591.

Response: In the first version of the manuscript there was an extensive presentation of some research results on will and volition. Following the reviewer's comments on this version, the introduction was rewritten and focused on the specific construct on which our scale was developed. However, we agree with the new reviewer's suggestion and have now deleted philosophical and historical concepts, but mentioned some theories on will and the related scales in the Introduction section. Finally, we apologize but we cannot understand how to include the mentioned “Recommended papers”.

Step 1: 

A critical ratio analysis is necessary to demonstrate the item discriminability for each question.

Response: We are not sure we can understand this comment. In Step 1 we are testing content validity, and we asked some experts to rate the item relevance on a 3-point scale (from 1=no relevance to 3=high relevance) and to assign them the dimension to which they believed each item pertained. The results are reported in the “Content Validity” paragraph. Item discriminability is reported in Study 2 and Study 3 as item-total correlations.

Step 2:

The sample exhibits a gender imbalance that needs addressing.

Response: We deem that 66% of female participants do not represent a biased gender distribution.

The reported degrees of freedom (df) value for Bartlett’s test of sphericity appears unusual and should be verified.

Response: The value was checked and confirmed.

Step 3:

The rationale for item deletion needs to be more comprehensive to justify the decisions made during this step.

Response: In the “CFA and Reliability” section of Study 3, we explained that we aimed to shorten and strengthen the scale. Thus, we decided to drop one item with a low factor loading, and items that did not contribute to the internal consistency of the respective scale. We would like to point out that Item selection is a complex decision-making process involving selecting the most suitable items from a range of available possible solutions linked to the adoption of more or less restrictive selection criteria. In the paper, we made explicit the criteria we adopted to develop this scale.

---

## [Editor Report · Decision Letter 3]

31 May 2024

Measuring Strong, Skillful, Good and Transpersonal Will: The development of the Multidimensional Will Scale

PONE-D-23-06766R3

Dear Dr. Bonacchi,

We’re pleased to inform you that your manuscript has been judged scientifically suitable for publication and will be formally accepted for publication once it meets all outstanding technical requirements.

Kind regards,

Yansong Li

Academic Editor

PLOS ONE